# Advancing Surgical Arrhythmia Ablation: Novel Insights on 3D Printing Applications and Two Biocompatible Materials

**DOI:** 10.3390/biomedicines12040869

**Published:** 2024-04-15

**Authors:** Cinzia Monaco, Rani Kronenberger, Giacomo Talevi, Luigi Pannone, Ida Anna Cappello, Mara Candelari, Robbert Ramak, Domenico Giovanni Della Rocca, Edoardo Bori, Herman Terryn, Kitty Baert, Priya Laha, Ahmet Krasniqi, Ali Gharaviri, Gezim Bala, Gian Battista Chierchia, Mark La Meir, Bernardo Innocenti, Carlo de Asmundis

**Affiliations:** 1Heart Rhythm Management Centre, Postgraduate Program in Cardiac Electrophysiology and Pacing, European Reference Networks Guard-Heart, Vrije Universiteit Brussel, Universitair Ziekenhuis Brussel, 1050 Brussels, Belgium; cinziamonaco90@gmail.com (C.M.);; 2Cardiac Surgery Department, Vrije Universiteit Brussel, Universitair Ziekenhuis Brussel, 1050 Brussels, Belgium; rani.kronenberger@uzbrussel.be (R.K.);; 3BEAMS Department, Bio Electro and Mechanical Systems, École Polytechnique de Bruxelles, Université Libre de Bruxelles, 1050 Brussels, Belgiumbernardo.innocenti@ulb.be (B.I.); 4Research Group Electrochemical and Surface Engineering (SURF), Department Materials and Chemistry, Vrije Universiteit Brussel, Universitair Ziekenhuis Brussel, 1050 Brussels, Belgium; 5In Vivo Cellular and Molecular Imaging Laboratory, Vrije Universiteit Brussel, Universitair Ziekenhuis Brussel, 1050 Brussels, Belgium

**Keywords:** ablation, additive manufacturing, material testing, cardiac, epicardial, three-dimensional printing (3D printing)

## Abstract

To date, studies assessing the safety profile of 3D printing materials for application in cardiac ablation are sparse. Our aim is to evaluate the safety and feasibility of two biocompatible 3D printing materials, investigating their potential use for intra-procedural guides to navigate surgical cardiac arrhythmia ablation. Herein, we 3D printed various prototypes in varying thicknesses (0.8 mm–3 mm) using a resin (MED625FLX) and a thermoplastic polyurethane elastomer (TPU95A). Geometrical testing was performed to assess the material properties pre- and post-sterilization. Furthermore, we investigated the thermal propagation behavior beneath the 3D printing materials during cryo-energy and radiofrequency ablation using an in vitro wet-lab setup. Moreover, electron microscopy and Raman spectroscopy were performed on biological tissue that had been exposed to the 3D printing materials to assess microparticle release. Post-sterilization assessments revealed that MED625FLX at thicknesses of 1 mm, 2.5 mm, and 3 mm, along with TPU95A at 1 mm and 2.5 mm, maintained geometrical integrity. Thermal analysis revealed that material type, energy source, and their factorial combination with distance from the energy source significantly influenced the temperatures beneath the 3D-printed material. Electron microscopy revealed traces of nitrogen and sulfur underneath the MED625FLX prints (1 mm, 2.5 mm) after cryo-ablation exposure. The other samples were uncontaminated. While Raman spectroscopy did not detect material release, further research is warranted to better understand these findings for application in clinical settings.

## 1. Introduction

Three-dimensional printing (3DP) is gaining increasing recognition in interventional cardiology and cardiac surgery [1,2]. It has proven a valuable tool for surgical planning, simulation training, intraoperative guidance, and customized device printing [3]. Thus far, the use of 3DP for intraoperative guides during the ablation of arrhythmogenic substrates requires further exploration. While its feasibility has been established [4,5], there is a paucity of data concerning the safety and integrity of 3D printing materials in proximity to (cryo-)thermal ablative energy.

In this study, we conducted a comprehensive analysis of two 3DP materials—a resin (MED625FLX; Stratasys, MN, USA) and a thermoplastic polyurethane elastomer (TPU95A; Ultimaker, The Netherlands, NLD)—to assess the potential suitability for 3DP guide prototyping. We developed an in vitro model to assess three hypotheses: (1) the two biocompatible materials, already available on the market, retain their physical properties after standard sterilization, thus allowing safe introduction into a sterile field; (2) the use of 3DP models during energy-intensive procedures does not adversely impact the energy propagation or temperatures of the biological tissue directly beneath the 3DP material; (3) the application of energy sources used in intraoperative settings, such as cryothermal (CRYO) or radiofrequency (RF) energy, does not compromise the model, nor does it emit material particles onto the underlying tissue.

## 2. Materials and Methods

This in vitro study aimed to mimic clinical epicardial ablation to examine the behavior of 3DP materials in response to the proximity of ablation energy delivery. The study entailed the following steps: (1) the selection of 3DP materials and the printing process; (2) the sterilization and assessment of post-sterilization geometrical properties using various designs; (3) conducting in vitro ablation experiments on semicircle 3D-printed prototypes in a wet lab; and (4) a post-experimental analysis to assess material release.

### 2.1. Selection of 3D Printing Material

A variety of biocompatible materials designed for 3D printing are available on the market, with abundant applicability in medical settings [6]. The 3DP materials in this study, however, were required to meet several technical specifications, including (1) compliance with the ISO 10993-1 biocompatibility criteria [7]; (2) the ability to maintain geometrical integrity after sterilization, without material degradation; (3) elastic properties suitable for application on the cardiac surface; and (4) resistance to energy source applications (CRYO, unipolar RF, bipolar RF). Moreover, the materials were required to be compatible with the 3D printer technologies available at our center: PolyJet (Objet260 Connex1, Stratasys, MN, USA) or Fused Deposition Modeling (Mega Zero, Anycubic, Hongkong, HK). MED625FLX and TPU95A are known for their biomedical applications, as reported in the literature and manufacturer datasheets [8,9,10], and met these criteria. The workflow from imaging to 3D printing has been detailed in previous literature [4,5,11].

#### Material Sterilization and Geometry Testing

The sterilization process is a critical step to consider when choosing appropriate 3DP materials for sterile medical procedures. Various sterilization methods are documented in the existing literature [12,13]. For our experiments, we employed Vaporized Hydrogen Peroxide (VHP) sterilization using STERRAD NX (Johnson & Johnson, Irvine, CA, USA), as previously described [14].

Geometry testing aimed to verify material integrity after sterilization by comparing each model macroscopically before and after the sterilization process. In addition, the surface finish was evaluated to prevent damage to fragile epicardial tissue. Multiple diverse designs were 3D printed at thicknesses of 0.8 mm, 1 mm, 2.5 mm, and 3 mm in MED625FLX and TPU95A, designed to closely match the contours of the epicardial surface. For each design, we drew upon previously published proofs-of-concept, including models for ventricular tachycardia scar ablation, Brugada Syndrome ablation, and coronary artery bypass grafting [4,5,11]. Each model was positioned on a non-linear rigid support, simulating the cardiac surface, which was also generated from CT scan reconstructions. Boundaries on the rigid support were measured before and after sterilization to verify geometrical integrity. Macroscopic variations in geometry were assessed by our engineering team using a predefined scale, where 0 indicated no visible material damage, with the geometry remaining intact and unchanged; 1 indicated no visible material damage, with the geometry remaining intact but with minor measurable differences; and 2 indicated visible damage, with the geometry significantly altered and fragmented [4].

### 2.2. In Vitro Ablation Experiment

The ablation experiment was conducted in a wet-lab setup [4,15]. The aim was to verify the intra-procedural safety of both materials by assessing their energy propagation behavior and, hence, whether ablation energy would extend underneath the 3DP material. Furthermore, following the experiments, the underlying tissue was examined to investigate the potential release of 3DP material particles.

To prevent uneven energy distribution caused by the 3DP geometry, we opted for a standard semicircle layout, featuring a central eyelet designated for energy application (Figure 1). These prototypes were designed with two internal measurement points (distances of 1 mm and 11 mm from the ablation energy source), providing designated locations for the insertion of two temperature sensors (Voltcraft thermometer PL-125-T4 with 4 thermocouples type K). This enabled real-time temperature monitoring of the biological tissue immediately underneath the material during energy delivery. The semicircles were printed using thicknesses of 1 mm and 2.5 mm, taking into account the thicknesses that previously passed the geometry test (as detailed in the Section 3).

### 2.3. Type of Energy Delivery

Three distinct ablation energy sources were used, including cryo-energy (10 cm aluminum CryoICE Cryoablation Probe, AtriCure Inc., West Chester, OH, USA), unipolar radiofrequency (uni-RF) (3.5 mm tip ablation catheter, FlexAbility, St. Jude Medical; Diathermy pencil 320 cm, Fiab SPA, Florence, Italy), and bipolar radiofrequency energy (bi-RF) (unidirectional RF linear device, Coolrail, AtriCure Inc., West Chester, OH; Diathermy pencil 320 cm, Fiab SPA, Florence, Italy). It was already reported that altering ablation settings such as power, catheter direction, irrigation rate, and irrigation saline temperature while maintaining the same ablation index does not produce notable differences in lesion size [16]. The ablation parameters were adjusted to replicate clinical settings and kept constant throughout each conducted measurement (CRYO: −70 °C, 120 s; uni-RF ablation catheter: 50 W, 70 °C, 60 s; uni-RF electrocautery: 50 W, 30 s; bi-RF ablation catheter: 70 W, 70 °C, 60 s; bi-RF electrocautery: 50 W, 30 s). The same catheters were used for the whole duration of the experiment.

### 2.4. Experimental Setup

The entire experiment was performed on ex vivo porcine tissue samples (breed: Belgian Landrace) with a thickness of 10.0 mm; the reason is that the surgical guide is aimed at ventricles and the porcine muscle thickness is approximatively similar to the human ventricular wall [17]. There was no need for approval from the Institutional Animal Welfare Committee for ex vivo tissue utilization. To reproduce human physiological conditions, the porcine tissue samples were submerged in a plexiglass box filled with physiological saline solution and maintained at a constant temperature (37 °C, physiological body temperature) using an immersion thermostat. After reaching the desired temperature, the tissue samples were removed from the container and positioned onto an external platform. The temperature of the samples was controlled using a CJ101 Digital Probe Thermometer. The catheter probe tip was positioned parallel to the myocardial tissue surface, within the designated eyelet. Throughout the energy delivery process, the ablation parameters were monitored and kept constant. Temperatures were measured at regular time intervals during the ablation: 0, 10, 30, and 60 s.

### 2.5. Post-Procedural Tissue Analysis

Following the in vitro experiment, tissue samples were excised to assess the potential presence of 3DP material particle release. The tissue underwent initial freezing, followed by fixation in formaldehyde 4%. The superficial layer was sectioned using a microtome (5–10 µm), and placed on glass slides. Subsequently, a combined analysis was performed, including scanning electron microscopy with energy-dispersive X-ray (SEM/EDX; JEOL JSM-IT300 Scanning Electron Microscope, coupled with an Oxford Instruments EDS detector for X-ray analysis, with an energy resolution of 127 eV). Backscattered electron imaging was used to visualize areas containing heavier atoms. Secondly, Raman spectroscopy (LabRAM HR Evolution—Horiba Scientific; green laser 532 nm–max., 1 mW objective 50x) was conducted, with particular attention paid to tissues with potential contamination in electron microscopy (samples that came into contact with 3DP material), as well as the control samples (ablated porcine tissue without the presence of 3DP material) [18]. The spectra obtained from the control samples were analyzed and used as a reference for the normal composition of biological samples. The same analysis was conducted individually for each 3DP material. Comparing the spectroscopic composition of biological material with that of 3D-printed material unveiled potential post-ablation particle release [19].

### 2.6. Statistical Analysis

All variables were tested for normality with the Shapiro–Wilk test and for homoscedasticity of the model residual with the Levene test. Descriptive statistics were calculated for variables of interest. The significance of the relationships between temperature reached under the 3D prints during ablation (dependent variable) and the type of ablation energy, distance from the ablation source, ablation time, and thickness and type of material was assessed by means of multivariate analysis of covariance (mixed ANCOVA and Dunn’s post hoc comparisons). Significant differences were further evaluated by using a Paired Samples *T*-Test. Values of *p* < 0.05 were considered statistically significant. Statistical analysis was performed by using R software version 3.6.2 (R Foundation for Statistical Computing, Vienna, Austria).

## 3. Results

### 3.1. Geometry Test after Sterilization

Multiple 3DP prototypes with various designs were examined according to the aforementioned criteria (Figure 2). The geometrical integrity assessment revealed that both materials with the lowest thickness (0.8 mm) attained a score of 2. Moreover, materials with thicknesses of 1 mm and 2.5 mm achieved a score of 0, demonstrating durability through sterilization. TPU95A at 3 mm achieved a score of 1, whereas MED625FLX at 3 mm scored 0. Detailed results are presented in Table 1. Furthermore, there were no apparent surface irregularities observed on any of the 3D prints.

### 3.2. In Vitro Ablation

Table 2 provides a quantitative overview of the ablation experiment. A total of one hundred distinct ablation points were delivered to samples of porcine tissue using different ablation energy sources (Figure 3). Initially, applications were performed on control tissue without 3D-printed material. Following this, applications were performed in proximity to MED625FLX and TPU95A, respectively. The ANCOVA analysis (Table 3) revealed a significant effect of the independent variables on the dependent variable (temperature on porcine tissue). Specifically, the type of 3DP material, the ablation energy source, and their factorial combinations with distance had significant impacts on temperature (Temperature*Material, *p* = 0.003; Temperature*Ablator, *p* < 0.001; Temperature*Material*Distance, *p* = 0.02; Temperature* Distance *Ablator, *p* < 0.001). In the post hoc test using Dunn’s procedure, temperature was found to be significantly influenced by distance and the type of ablation energy (Temperature* Distance, *p* < 0.001; Temperature*Ablator*Cryo-biRF, *p* < 0.001; Temperature*Ablator*Cryo-uniRF, *p* < 0.001; Temperature*Ablator*biRF-uniRF, *p* < 0.001) (Table 4).

### 3.3. Electron Microscopy and Raman Spectroscopy

SEM/EDX revealed that the main components detected in the control samples were carbonium (C) and oxygen (O) (Figure 4A,B).

No nitrogen (N) or sulfur (S) was detected in the control group; they were instead highly present in MED625FLX and TPU95A (Figure 5 and Figure 6).

Electron microscopy revealed traces of N and S in the biological samples collected underneath MED625FLX 3DP after cryo-ablation. This was observed in samples with thicknesses of both 1 mm and 2.5 mm (Figure 7 and Figure 8).

The other biological tissue collected below MED625FLX 3DP after RF applications, and all biological tissue collected below TPU95A, appeared uncontaminated. Raman spectroscopy was further performed, and comparing the spectroscopic composition of the biological material with that of the 3D-printed material unveiled potential post-ablation particle release (Figure 9). Particular attention was paid to the tissue with contamination in electron microscopy, which did not detect any observable evidence of 3DP material release after ablation, specifically, particles larger than 1 μm. The remaining analyzed samples exhibited no evidence of elements other than those found in the control sample.

## 4. Discussion

The emergence of three-dimensional printed models has created new opportunities in the evolving field of precision medicine and personalized surgery. By leveraging 3D-printed models generated from integrated imaging and mapping data, we can effectively improve the efficacy and safety of surgical arrhythmia procedures. However, the lack of prior research leaves uncertainty regarding the selection of 3DP materials for these endpoints, as no other research groups have provided data on their safety and suitability. Building upon our previous investigations into material testing [14,15], we explored various designs in our printing process to evaluate their responses to sterilization procedures. Furthermore, we conducted additional microscopic assessments and temperature analyses to deepen our understanding of the safety profile of MED625FLX and TPU95A for 3DP guide prototyping in arrhythmia ablation.

Our findings illustrate that MED625FLX and TPU95A can be printed and sterilized while retaining their shape across thicknesses of 1.0 mm and 2.5 mm. MED625FLX at 3 mm can also be achieved; however, there exists a delicate equilibrium between the dimensional stability required for this process and the flexibility necessary for minimally invasive purposes. Thin forms measuring 0.8 mm should therefore not be used. This validates previous testing conducted to evaluate geometric integrity following ablation [14]. Additionally, it highlights unsuccessful outcomes with the use of 0.8 mm.

Furthermore, thermal statistical analysis was conducted to elucidate the relationships between temperature and multiple variables. This analysis revealed that the type of material, the energy source, and their factorial combination with distance were significant factors influencing temperatures beneath the 3DP material. Post hoc tests indicated significant effects of distance and energy type, with no notable temperature differences based on the material and its thickness. Moreover, Dunn’s post hoc comparison showed a higher temperature rise with unipolar ablation compared with the other sources.

Electron microscopy and Raman spectroscopy were used to verify the absence of (micro)particle release from the 3DP material onto the biological tissue after ablation. Electron microscopy revealed particles of nitrogen and sulfur in post-cryo-ablation tissue underneath MED625FLX (1 mm; 2.5 mm), suggesting potential material deposition on the tissue surface after ablation. Raman spectroscopy, however, did not detect any observable evidence of MED625FLX material release after ablation, specifically, particles larger than 1 μm. Correlating these results with each other suggests that either the deposited 3DP material may be below the detection limit of Raman Spectroscopy, or these findings are related to other instruments. This highlights the need for complementary analyses like X-ray photoelectron spectroscopy or Fourier transform infrared spectroscopy.

### Limitations

Taken together, the safety conclusions drawn based on this dataset have to be taken with caution, as this preclinical study faces some constraints. Firstly, conducting in vitro experiments on ex vivo porcine tissue fails to capture the complexity of physiological responses in a beating heart, hindering the accurate assessment of temperature variations during ablation. Therefore, the next phase should involve in vivo studies in swine, including assessing the viability and proliferation of porcine cardiomyocytes when cultured on selected material, live/dead imaging tests, and immunofluorescence, to morphologically evaluate the cells on materials before and after ablation treatment. Secondly, due to the simplified nature of the experiments and the limited data available, larger studies are necessary to provide more comprehensive insights. Additional assessments, such as examining microstructural changes in the 3D printing material and conducting material analyses, are necessary to ensure a comprehensive understanding of the material’s safety implications. As this research is part of a larger data collection effort, we are currently investigating microstructural changes in 3D printing materials in response to sterilization and ablation procedures.

## 5. Conclusions

This study contributes to improving our understanding of safe and suitable 3D-printed tools for ablation procedures. In conclusion, our study shows that MED625FLX and TPU95A can effectively be printed and sterilized while maintaining their shape across thicknesses of 1.0 mm and 2.5 mm, except for thinner forms (0.8 mm), which should be avoided. Thermal analysis revealed that material type, energy source, and their factorial combination with position significantly influence temperatures beneath the 3D-printed material. Additionally, electron microscopy revealed traces of N and S underneath MED625FLX 3DP prints with thicknesses of 1 mm and 2.5 mm after exposure to cryo-ablation, whereas the other samples appeared uncontaminated. While Raman spectroscopy did not detect significant material release after ablation, further microscopic testing is needed to clarify these findings. Further research is essential to validate and refine these findings for practical application in clinical settings.

## Figures and Tables

**Figure 1 biomedicines-12-00869-f001:**
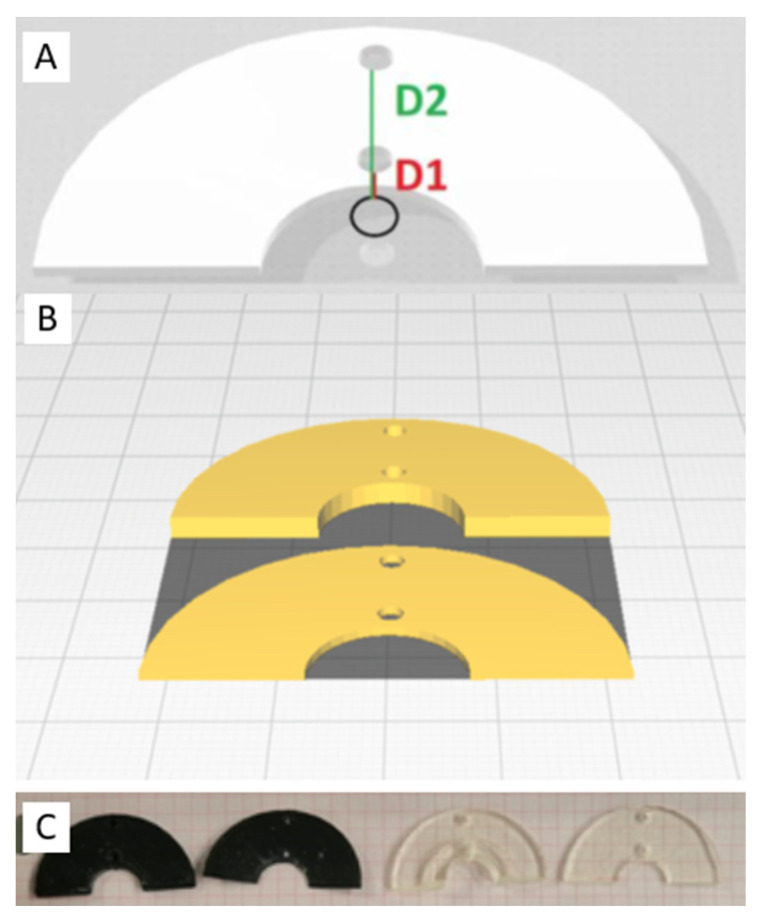
3D-printed prototypes for in vitro ablation experiments: (**A**) An eyelet for the ablation catheter (black circle) placed centrally. Two 1 mm internal lumens were created for thermocouple insertion at distances of 1 mm (D1) and 11 mm (D2) from the energy source. (**B**) Computerized blender samples for 3D printer. (**C**) 3DP samples, with different thicknesses and materials (TPU: black; MED625FLX: transparent).

**Figure 2 biomedicines-12-00869-f002:**
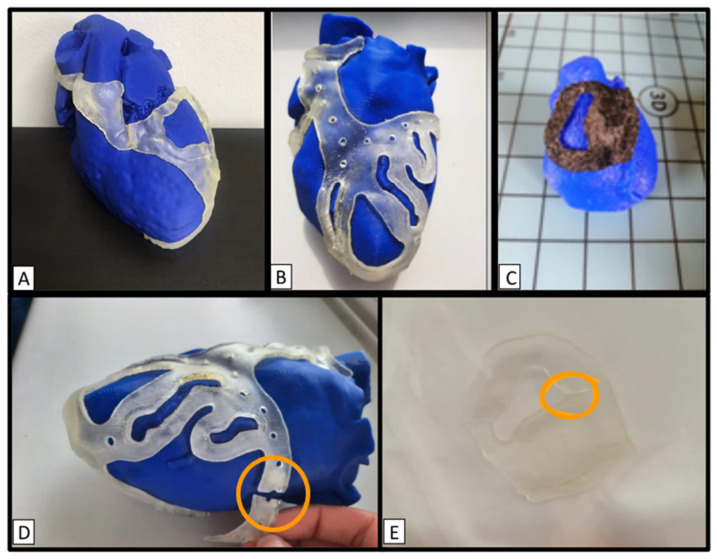
Assessment of geometric integrity of various 3D-printed designs before and after sterilization. (**A**–**C**) Pre-sterilization models; (**D**,**E**) two failed post-sterilization designs (orange circles highlight the areas of damage). (**A**) a 3D-printed guide of ventricular tachycardia-inducing scar substrate located in the left ventricle, 3D printed in MED625FLX (2.5 mm thickness); (**B**) 3D-printed model revealing coronary stenosis in a NSTEMI patient, printed in MED625FLX (3 mm thickness) [8]; (**C**) 3D-printed model of arrhythmogenic substrate in Brugada Syndrome (TPU; 1 mm); (**D**) example of material breakage (orange circle) after sterilization on the coronary model shown in panel (**B**); (**E**) area of damage (orange circle) following sterilization on a Brugada Syndrome model in MED625FLX (0.8 mm thickness).

**Figure 3 biomedicines-12-00869-f003:**
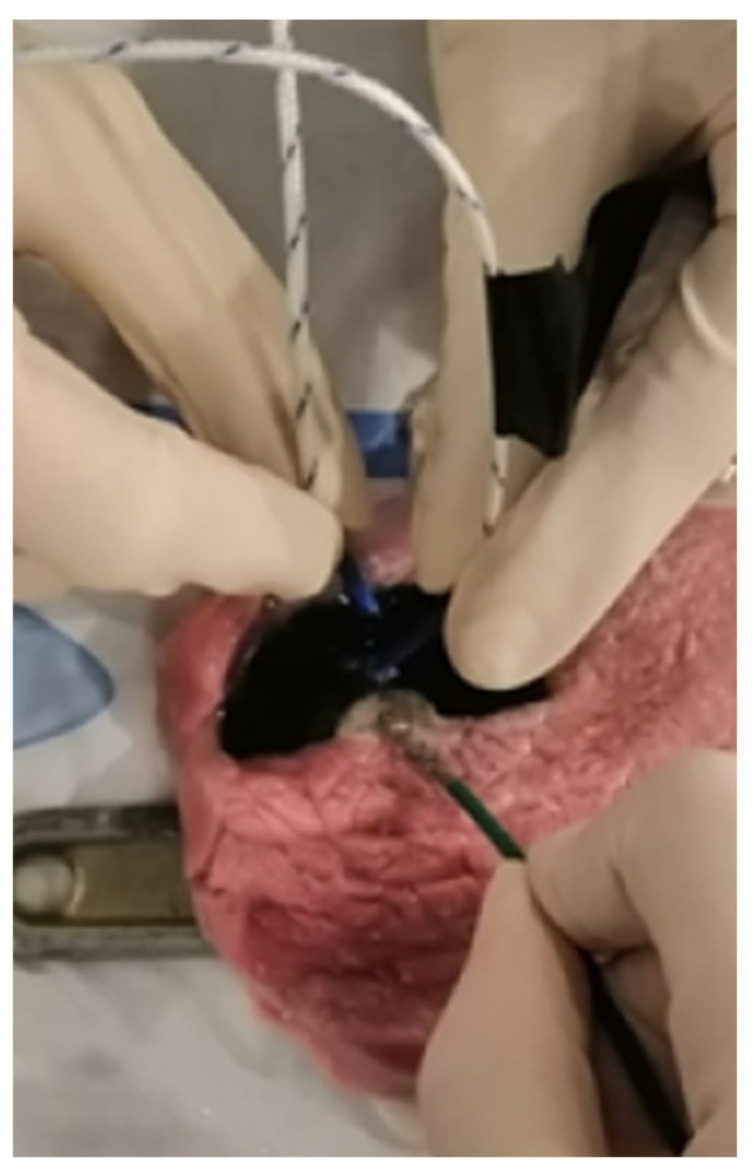
A segment of the wet-lab experiment. The ablation catheter is placed in the central eyelet of the 3DP prototype. The temperature beneath the 3DP material is measured during the ablation using thermocouples, at distances of 1 mm and 11 mm from the energy source.

**Figure 4 biomedicines-12-00869-f004:**
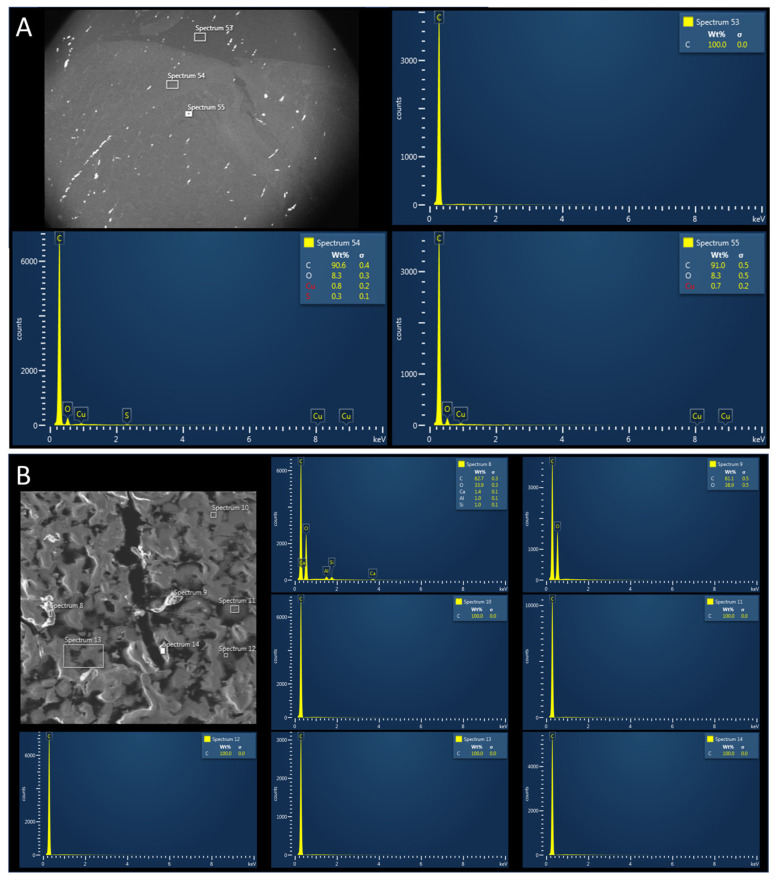
(**A**) Scanning electron microscopy with energy-dispersive X-ray showing control tissue samples (ablated porcine tissue without 3D printing material). The main components detected include carbonium (C) and oxygen (O). (**B**) Higher magnification confirming that the main components detected in the three measurements include carbonium (C) and oxygen (O).

**Figure 5 biomedicines-12-00869-f005:**
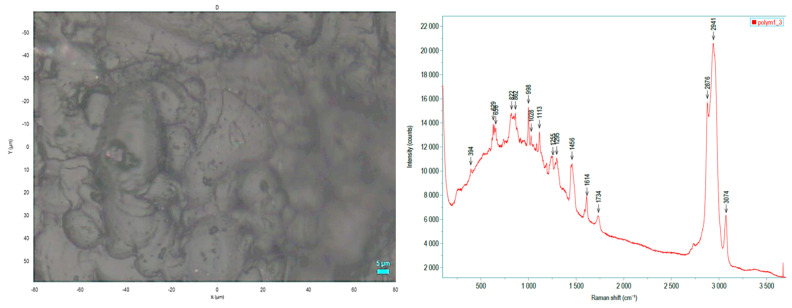
**Left** panel: 3D print in MED625FLX was assessed using an electron microscope. **Right** panel: Raman spectrum measured centrally (4 measurements). The same spectrum was obtained for all measured positions, and no material damage (due to the laser irradiation (max. power) was observed.

**Figure 6 biomedicines-12-00869-f006:**
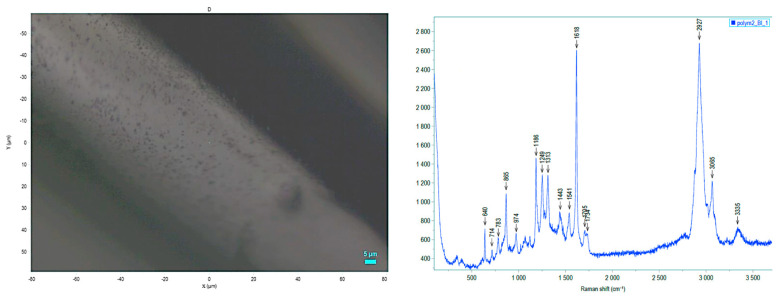
**Left** panel: 3D print in TPU95A was evaluated using an electron microscope. **Right** panel: Raman spectrum was measured centrally (4 measurements). The same spectrum was obtained for all measured positions; at a high laser power (>1 mW), the material was damaged in the irradiated area, compatible with the information reported on the datasheet. Laser power < 1 mW used for the analysis.

**Figure 7 biomedicines-12-00869-f007:**
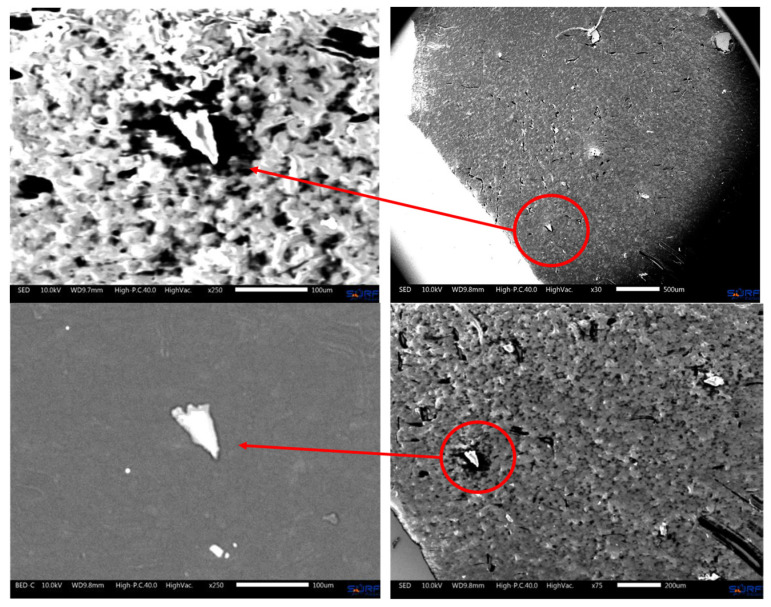
Images obtained, through an electron microscope, of biological tissue beneath MED625FLX prototypes (1 mm; 2.5 mm) after cryo-ablation. Particles of nitrogen and sulfur were identified (EDX) (red circles), which are not typically present in biological tissue.

**Figure 8 biomedicines-12-00869-f008:**
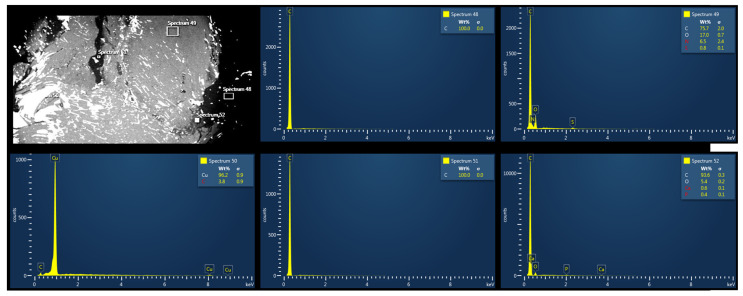
Scanning electron microscopy with energy-dispersive X-ray spectroscopy of biological tissue below MED625FLX material after cryo-ablation. Traces of nitrogen were detected.

**Figure 9 biomedicines-12-00869-f009:**
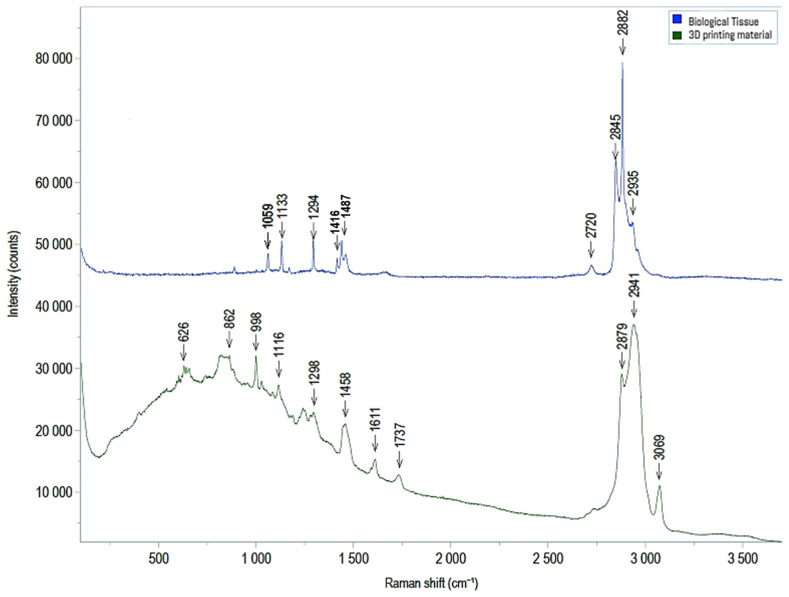
Comparison between Raman spectroscopy results for reference tissue sample (blue) and MED625FLX material (green).

**Table 1 biomedicines-12-00869-t001:** Geometry testing of 3D-printed prototypes after sterilization using a predefined scale; 0 = no visible material damage, with the geometry remaining intact and unchanged; 1 = no visible material damage, with the geometry remaining intact but with minor measurable differences measured on the rigid support pre- and post-sterilization; 2 = visible damage, with the geometry significantly altered and fragmented.

3D Printing Material	Thickness (mm)	Geometry Damage Index
MED625FLX	0.8	2
	1	0
	2.5	0
	3	0
TPU95A	0.8	2
	1	0
	2.5	0
	3	1

**Table 2 biomedicines-12-00869-t002:** Overview of energy-delivery applications on the biological tissue using cryo- and radiofrequency energy sources.

	Thickness of 3DP Model (mm)	Cryo N. Applications Delivered	Uni-RF N. Applications Delivered		Bi-RF N. Applications Delivered		N. Applications/ Material
		Ablation catheter	Ablation catheter	Electro-cautery	Ablation catheter	Electro-cautery	
Control	N/A	4	4	4	4	4	20
MED625FLX	1	4	4	4	4	4	20
	2.5	4	4	4	4	4	20
TPU	1	4	4	4	4	4	20
	2.5	4	4	4	4	4	20
N. Applications/energy type 20			40		40		100

3DP: 3D-printed; Cryo: cryo-energy; Bi-RF: bipolar radiofrequency energy; Uni-RF: unipolar radiofrequency energy; N/A: not applicable.

**Table 3 biomedicines-12-00869-t003:** ANCOVA analysis. Statistically significant findings are highlighted in bold.

Mixed ANCOVA—Temperature					
Cases	Sum of Squares	DF	Mean Square	F Statistic	*p*-Value
Thickness	15.969	1	15.969	0.366	0.545
Material	378.539	1	378.539	8.681	0.003
Distance	2.260	1	2.260	0.052	0.820
Energy	55,706.124	2	27,853.062	638.751	<0.001
Time	75.776	3	25.259	0.579	0.629
Thickness ✻ Material	8.724	1	8.724	0.200	0.655
Thickness ✻ Distance	140.150	1	140.150	3.214	0.074
Thickness ✻ Energy	67.475	2	33.737	0.774	0.462
Thickness ✻ Time	9.467	3	3.156	0.072	0.975
Material ✻ Distance	238.805	1	238.805	5.476	0.020
Material ✻ Energy	77.657	2	38.828	0.890	0.411
Material ✻ Time	20.792	3	6.931	0.159	0.924
Distance ✻ Energy	22,681.436	2	11,340.718	260.075	<0.001
Distance ✻ Time	14.797	3	4.932	0.113	0.952
Energy ✻ Time	522.313	6	87.052	1.996	0.065
Thickness ✻ Material ✻ Distance	0.665	1	0.665	0.015	0.902
Thickness ✻ Material ✻ Energy	140.044	2	70.022	1.606	0.202
Thickness ✻ Material ✻ Time	3.036	3	1.012	0.023	0.995
Thickness ✻ Distance ✻ Energy	125.124	2	62.562	1.435	0.240
Thickness ✻ Distance ✻ Time	0.709	3	0.236	0.005	0.999
Thickness ✻ Energy ✻ Time	37.848	6	6.308	0.145	0.990
Material ✻ Distance ✻ Energy	33.523	2	16.762	0.384	0.681
Material ✻ Distance ✻ Time	37.493	3	12.498	0.287	0.835
Material ✻ Energy ✻ Time	7.003	6	1.167	0.027	1.000
Distance ✻ Energy ✻ Time	425.703	6	70.950	1.627	0.139
Thickness ✻ Material ✻ Distance ✻ Energy	33.727	2	16.864	0.387	0.680
Thickness ✻ Material ✻ Distance ✻ Time	12.126	3	4.042	0.093	0.964
Thickness ✻ Material ✻ Energy ✻ Time	2.013	6	0.335	0.008	1.000
Thickness ✻ Distance ✻ Energy ✻ Time	1.027	6	0.171	0.004	1.000
Material ✻ Distance ✻ Energy ✻ Time	10.228	6	1.705	0.039	1.000
Thickness ✻ Material ✻ Distance ✻ Energy ✻ Time	8.531	6	1.422	0.033	1.000
Residuals	15,567.165	357	43.606		

Note: Type III Sum of Squares; DF: degrees of freedom.

**Table 4 biomedicines-12-00869-t004:** Dunn’s post hoc comparison. Statistically significant findings are highlighted in bold.

**Temperature ***			z-test	W_i_	W_j_	*p*-value
	Distance (mm)	1–11	3.764	250.000	203.799	**<0.001**
	Energy type	Cryo–bi-RF	−12.020	78.378	259.172	**<0.001**
		Cryo–uni-RF	−17.711	78.378	345.217	**<0.001**
		Bi-RF–uni-RF	−5.702	259.172	345.217	**<0.001**
	Material	MED625FLX-TPU	−0.645	222.828	230.769	0.260
	Thickness (mm)	1–2.5	1.369	235.858	218.998	0.086

## Data Availability

The data presented in this study are available on request from the corresponding author.

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
