# Peer review of "Advancing Surgical Arrhythmia Ablation: Novel Insights on 3D Printing Applications and Two Biocompatible Materials"

_biomedicines, 2024, doi:10.3390/biomedicines12040869_

Round 1

Reviewer 1 Report

Comments and Suggestions for Authors

The manuscript is well presented and the purpose is well defined. It comes slightly out of my expertise, more biological than physical. However, I believe that for a journal like Biomedicine it is necessary to add some biological evidence, the manuscript seems to me much more physical than biological. The authors have already highlighted the strengths (which are many and I appreciate) and the critical points of their research, however I suggest expanding the plethora of biological investigations on the biocompatibility of printed materials necessary for publication in a journal such as Biomedicine. Biocompatibility testing of cardiomyocytes (heart muscle cells) in the context of 3D printing is crucial for the development of functional and safe tissue engineering constructs or biomedical devices. Otherwise I would suggest the authors change journal (like Bioprinting). 

Please provide at least for this paper cell viability and proliferation test, as ISO 10993 standards provide guidelines for cytotoxicity and viability testing.  

- Asses the viability and proliferation of porcine cardiomyocytes  when cultered on selected material at different time points and after printing;

- provide Live/dead imaging tests or IF, to evaluate morphologically the cells on materials and after ablation treatment. 

Comments on the Quality of English Language

Minor editing of English language required

Reviewer 2 Report

Comments and Suggestions for Authors

 in the title should be added "of two biocompatible  3D printing materials "

 in the lines 57, 84-85, 94, the country of origin should be added in the brakets.

line 78 please add the relevant reference and in 2.1 paragraph in general (1-2-3-4, lines 79-83).

line 93 please add the relevant references

 in the paragraph lines 106-110 at the end please add the relevant references. 

in line 113 please provide the relevant reference for the set up

lines 141 143. in the text there is a mentions about studies and at the end there is only one reference. please add the others.

151 please mention how, where and under which permission did you obtain the swine tissue

151 please add the breed of the pig and the relevant reference where your experiment set up was based

 in the methods and materials please define whether you used the same catheter or different, how many times and the relevant references.

 As a general comments the paper needs more references to be strength. 

Round 2

Reviewer 1 Report

Comments and Suggestions for Authors

Dear authors, 

I thank you for the response to my criticisms and I perfectly understand the lack of time to perform the experiments requested.

Although I believe this paper should be published in a more physical journal, I accept the manuscript.

Author Response

Dear rewiever,

Thank you very much for taking the time to review this manuscript. 

Best regards

Reviewer 2 Report

Comments and Suggestions for Authors

please add the breed of the pig as requested

Author Response

Dear reviewer,

Thank you very much for taking the time to review this manuscript. The breed is added.

Best regards
